# Freeze–Thaw Durability of Basalt Fibre Reinforced Bio-Based Unsaturated Polyester Composite

**DOI:** 10.3390/ma16155411

**Published:** 2023-08-02

**Authors:** Abu T. Shahid, Mateus Hofmann, Mário Garrido, João R. Correia, Inês C. Rosa

**Affiliations:** 1Civil Engineering Research and Innovation for Sustainability (CERIS), Instituto Superior Técnico, Universidade de Lisboa, Av. Rovisco Pais 1, 1049-001 Lisbon, Portugal; mario.garrido@tecnico.ulisboa.pt (M.G.); joao.ramoa.correia@tecnico.ulisboa.pt (J.R.C.); ines.rosa@tecnico.ulisboa.pt (I.C.R.); 2Composite Construction Laboratory (CCLab), École Polytechnique Fédérale de Lausanne, BP 2124, Station 16, 1015 Lausanne, Switzerland; mateus.hofmann@epfl.ch

**Keywords:** bio-composite, wet freeze–thaw durability, bio-based unsaturated polyester resin, basalt fibre, mechanical properties, thermomechanical properties, gravimetric analysis, scanning electron microscopy

## Abstract

This paper presents an experimental study of the wet freeze–thaw (FT) durability of a fibre–polymer composite produced by vacuum infusion using an innovative bio-based unsaturated polyester resin (UPR) and basalt fibres. As the benchmark, an equivalent composite produced with a conventional (oil-based) UPR was also tested. The composites were preconditioned in water immersion for 30 days at 20 °C followed by exposure to wet FT for up to 300 cycles; each FT cycle consisted of 3 h in dry freezing condition (−20 °C) and 8 h in thawing condition (23 °C) submerged in water. The composites’ properties were assessed after preconditioning and after 100, 200, and 300 FT cycles, through mechanical (tensile, compressive, in-plane shear, interlaminar shear) and thermomechanical (dynamic mechanical analysis) tests. Gravimetric and scanning electron microscope analyses were also carried out. The results obtained show that the preconditioning stage, involving water immersion, caused most of the damage, with property reductions of 5% to 39% in the bio-composite, while in the oil-composite they ranged between 4% and 22%, being higher for matrix-dominated properties. On the other hand, FT alone had an insignificant effect on the degradation of material properties; after exposure to FT, property recovery was observed, specifically in matrix-dominated properties, such as interlaminar shear strength, which recovered by 12% in the bio-composite during exposure to FT. The overall performance of the bio-composite was inferior to the conventional one, especially during the preconditioning stage, and this was attributed to the hydrophilicity of some of the components of its bio-based resin.

## 1. Introduction

Fibre–polymer composites, also known as fibre reinforced polymer (FRP) composite materials, are increasingly finding their application in a wide range of industries, such as aerospace, transportation, sports equipment, and energy infrastructures [1]. Several properties of FRP materials make them attractive for specific applications, for instance, lightness, high strength, and durability under harsh environments [2]. Moreover, the cost reduction observed in recent years is also accelerating the use of composites in the price-sensitive construction sector as structural members, such as bridge decks or girders, and profiles of building frames, both in interior and exterior environments.

However, the production of fibre–polymer composites is heavily dependent on petroleum resources, as the polymer resins are produced from petroleum-derived chemical platforms. In addition, synthetic fibres, such as glass, which are predominantly used in low-cost composites, are obtained from very energy and resource-intensive processes [3]. The limited reserve of petroleum and price fluctuation in the petroleum market is exposing the composites industry to the risk of disruption in the supply of raw materials [4]. Current major global issues, such as global warming and climate change, are influencing both policymakers and industry to shift towards more sustainable and environmentally friendly materials [3].

To effectively achieve a sustainable shift in the composites industry, it is necessary to search for alternative materials that encompass lower environmental impacts compared to current conventional materials. One of the available routes for developing environmentally friendlier FRP composites is bio-based polymer resin derived from renewable resources and plant-based fibres. However, the limited mechanical properties and durability issues related to plant-based fibres still hinder their suitability for structural applications [5].

Basalt fibre is considered to be a high-performance inorganic ‘green’ material for its recyclability and lower energy production needs (compared to glass or carbon), as well as for offering non-toxic reactions with the environment, such as air and water [6]. The raw material is found abundantly in nature in the form of basalt rock, which is plentiful and encompasses 33% of the Earth’s crust [7]. Basalt fibres have been reported to offer better water and corrosion resistance compared to glass fibres [8,9,10,11]. Moreover, basalt fibres are compatible with conventional resins, such as unsaturated polyester, vinyl ester, and epoxy [12,13]. Therefore, the use of basalt-based fibre–polymer composites has been noticed in several sectors, such as geo-composites, industrial pipes, laminates and pre-pregs, basalt-lined castings, storage tanks, and clean energy and power grids, to name a few [1,14,15]. Basalt–epoxy composites have been found to outperform their glass–epoxy counterparts in terms of tensile, compressive, and flexural strength and modulus [16].

Another major component of fibre–polymer composites is the polymer matrix, which in most cases includes a thermoset resin, such as unsaturated polyester, vinyl ester, and epoxy; these resins are currently produced dominantly from petroleum-based chemical platforms. Recently, intensive research effort has been made to develop polymer resins fully or partially sourced from renewable resources [17,18,19,20,21]. The mechanical and thermomechanical performance of some of these bio-based polymer resins was found to be comparable with that of petroleum-based counterparts [21]. However, most of them lack the demonstration of their application in product development with fibre reinforcement. A notable exception can be found in the resin developed by Hoffman et al. [21], which has been used to produce glass fibre-based composites using the vacuum infusion method [22], and pultruded carbon fibre-based laminates [23]. However, there is a lack of studies regarding bio-based resins incorporating basalt fibres.

As fibre–polymer composites are now used in a wide range of sectors, the influence of environmental agents, for instance, elevated or low temperature, moisture, corrosive media, UV radiation, thermal cycling and freeze–thaw (FT) cycling on their durability (in terms of retention of mechanical, physical, and thermomechanical properties) is a major concern in many applications [24].

Existing literature shows that the damage caused by FT exposure to fibre–polymer composites can be higher than that caused by isothermal exposure at freezing temperatures [24,25]. In fact, the moisture absorbed in the structure of the composite material causes hydrolysis of the ester linkage, as well as plasticisation of the polymer matrix, resulting in an increase in the mobility of smaller molecular weight segments, deteriorating the performance of the composite material. Below freezing temperature, micro-cracks occur in the polymer matrix, which accelerate water absorption at higher temperatures, contributing to further hydrolysis and plasticisation of the polymer matrix. Moreover, the water accumulated in the cracks and voids expands in frozen conditions, exerting transverse stress and causing fibre–matrix debonding and growth in matrix cracking [25]. Therefore, FT durability together with infused moisture is a very important aspect to be assessed in a composite.

At the fibre level, the presence of water together with cyclic expansion and contraction due to FT cycles is also a source of degradation. Moisture causes the extraction of ions from the fibres and makes the environment basic, possibly causing pitting, leaching, and etching of the fibres surface, resulting in the degradation of their properties [26]. The metallic cations of basalt fibres can leach from the fibres surface through a reaction with hydroxyl ions of water. A corrosion shell of the silicate network is formed due to these erosions from the fibres surface, which eventually dissolves (etching) into the solution and accelerates the process by increasing the alkalinity of the in situ zone [26]. The damage caused in the polymer matrix together with fibre corrosion also results in the deterioration of the fibre–matrix interface, which acts as a conduit for the flow of corrosion medium, intensifying further damage [26].

FT studies on basalt–polymer composites are very limited. Although several studies [26,27,28,29] have investigated the effects of wet FT of basalt-based composites, none of them considered preconditioning in water before exposure to the FT cycles. Water immersion preconditioning is a very critical stage in FT studies (and it is prescribed in the standardised procedure indicated in ASTM D7792/D7792M [30]), allowing to simulate more realistically the damage mechanisms in a wet FT environment by ensuring that water occupies the porous space in the composite material. For instance, dry FT exposure can be expected to cause insignificant damage to composite materials [31], as the internal stresses developed through the FT cycle are typically much lower than the ultimate strength of the resin and fibre–matrix interface, thus not inducing micro-cracking [29]. However, in the presence of moisture, the damage to composite materials due to FT cycling is amplified [32]. As an example, preconditioned pultruded glass/polyester profiles subjected to wet FT experienced substantial reductions (13%) in flexural strength and an increase in flexural modulus (14%), while dry FT resulted in increases of 3% and 6%, respectively [31].

Among the studies available in the literature, fibre-dominated properties are found to be less significantly affected by FT exposure compared to matrix-dominated properties [28,29]. In the study by Shi et al. [27], wet FT exposure of basalt–epoxy composites caused almost no detrimental effect on longitudinal tensile strength (reduction of 1%) and modulus (10% increase). The damage tolerance of basalt fibre-based composites in wet FT has been found to be superior compared to glass fibre-based composites. Additionally, FT durability also depends on the type of polymer used in the matrix [28]. FT studies on bio-resin-based composites are also scarce. One study [33] investigated the wet FT durability of bio-resin-based fibre–polymer composites for 300 FT cycles (4 °C to −18 °C), consisting of epoxidised pine oil resin and furfural alcohol resin with glass and carbon fibre reinforcement; results showed a negligible effect of FT exposure in the tensile properties of the bio-based composites, which was comparable to the performance of a conventional epoxy-based counterpart.

One significant aspect that should be taken into account in the FT study of fibre–polymer composites is the temperature ranges at which the FT experiment is carried out. This feature has a complex relationship with damage mechanisms; FT exposure with a high upper bound temperature usually results in comparatively lower damage [32,34] compared to FT cycling with a low upper bound [29,35]. This might result from the post-curing phenomenon resulting from the exposure to higher temperatures.

Previous wet FT studies present significant differences as most of them did not follow any standards specific to fibre–polymer composites. This resulted in relevant differences in the number of FT cycles, duration of the cycles, temperature range, and immersion medium. Regarding the materials, a wide variety of resins, fibres, production techniques, and composite layups and thicknesses makes it almost impossible to compare the performance of the composites that were investigated in those studies. In addition, as mentioned, preconditioning is a critical part of wet FT assessment, and this procedure was not used in most studies. Furthermore, only a few properties were assessed to understand the damage caused by wet FT, which hinders the holistic analysis of the material’s performance in the environment. It is also worth mentioning that no study has been performed so far regarding the wet FT study of bio-resin-based basalt–polymer composite.

To overcome some of the gaps in the literature summarised above, this paper presents an experimental study of the wet FT durability of an innovative bio-resin-based basalt–polymer composite, which was assessed following the procedures outlined in the ASTM D7792/D7792M standard [30] and compared with an oil-based conventional counterpart with the same fibre architecture. The bio-based resin composition developed by Hoffman et al. [21] was used with slight modifications (detailed ahead). The damage caused by the preconditioning (immersion in water for 30 days) and from the FT cycles was assessed through changes in (i) mechanical properties—tensile strength and modulus, compressive strength, in-plane shear strength and modulus, and interlaminar shear strength, and (ii) thermophysical properties, particularly the glass transition temperature (T_g_), determined through dynamic mechanical analysis. In addition, gravimetric analyses were performed on the two types of composites to determine water uptake at 20 °C throughout a 60-day period and scanning electron microscope (SEM) observations were conducted on composite samples tested in tension before ageing, after preconditioning, and after the 300 FT cycles.

## 2. Materials

### 2.1. Bio-Based Resin

A partially bio-based unsaturated polyester resin (UPR) was used to produce a bio-based basalt–polymer composite. The resin development is detailed in [21]. Renewable di-acid monomer (fumaric acid) and diols (1,3-propanediol, isosorbide) were incorporated in the orthopthalic polyester pre-polymer chain (bio-based mass content of 84.1%, as per ISO 16620 [36]), followed by mixing with reactive diluents (RDs) (50 parts of partially bio-based unsaturated polyester pre-polymer to 50 parts of RDs), which resulted in an overall bio-based mass content of 42%. The composition of the main building block of pre-polymer includes phthalic anhydride (PA), fumaric acid (FA), 1,3-propanediol (PDO), and isosorbide (ISO), whereas the RDs consist of 2-hydroxy ethyl methacrylate (HEMA) (40% of total RDs) and styrene (a carcinogenic compound) (60% of total RDs), resulting in a reduction in styrene content compared to conventional UPRs. This 40:60 ratio of HEMA and styrene is the main difference between the resin used here and the formulation detailed in [21], for which this ratio was 50:50; this modification was introduced to improve the resin’s hydrolytic resistance following preliminary hygrothermal durability trials. The polymer matrix was cured catalytically by using peroxide (PMEK, 2%) together with octoate cobalt (1%) as the initiator. No additives were used.

### 2.2. Oil-Based Resin

For comparison, an equivalent (in terms of mechanical performance) orthotropic oil-based UPR was also considered, which was obtained from Scott Bader (Crystic U 904LVKTM, Wollaston, UK), containing orthopthalic acid in the pre-polymer. This resin was also cured in a catalytic process using PMEK (1%) and it also contained no additives.

Table 1 presents the mechanical and thermomechanical properties of the bio- and oil-based UPRs, namely their tensile and in-plane shear properties, and glass transition temperatures determined from the onset of the storage modulus (E′) decay and the peak of tan (δ), both obtained by dynamic mechanical analysis (DMA). Additionally, Table 1 provides the corresponding mean values, standard deviations, and coefficients of variation (where applicable), together with the standards that were followed, and the number of specimens tested to determine the properties. Overall, the mechanical and thermomechanical properties of the bio-based UPR are comparable to those of the oil-based commercial resin.

### 2.3. Basalt Fibre

Two types of basalt fibre mats, obtained from BASALTEX (Wevelgem, Belgium), were used: (a) unidirectional (0°) and (b) bi-directional (0°/90°). The areal weight of the unidirectional basalt fibre mat, which was stitched, was 550 g/m^2^ (0°: 500 g/m^2^ − 90°: 50 g/m^2^); while the bi-directional fibre mat was 940 g/m^2^ (50% in warp and 50% in the weft directions) and it was woven with a twill weave. The fibres were silane-sized with a content of 0.4–0.6%. The density of the un-sized filament is 2.67 kg/m^3^. In the fibre layup, a total of seven layers of basalt fibre mats were used, consisting of five layers of (0°) unidirectional (U) and two layers of (0°/90°) bidirectional (B) fibre mats arranged in the following sequence: (U/U/B/U/B/U/U).

### 2.4. Production of Composite Plates

Both bio-based and oil-based UPR were used to produce basalt–polymer composite plates using the vacuum infusion technique, which comprises the following three main stages: (i) preparation—stacking of the different mat layers over a metallic moulding plate (size of 1.25 m × 1.25 m), together with infusion meshes, peel ply and breather, and application of a plastic film (and other infusion accessories) and vacuum (pressure of about 40 mbar) for approximately 24 h; (ii) resin infusion, with visual observation of resin percolation and temperature increase (the vacuum system was connected for a period of 24 h); and (iii) demoulding. The composite laminates, 64 cm × 75 cm, were cured at room temperature for 48 h and then post-cured at 100 °C for 4 h. The nominal thickness of the plates was 4 mm, and the fibre mass content of the laminates was around 65%.

## 3. Methods

### 3.1. Freeze–Thaw Environment

The wet FT durability of the bio-UPR-based basalt–polymer composite and of its conventional counterpart was assessed according to the procedures outlined in the ASTM D7792/D7792M standard, using a freeze–thaw chamber from *Aralab*, model *FITOCLIMA 500 EPC45* (Figure 1).

The basalt–polymer composite plates were cut in a CNC router to dimensions 35 cm by 25 cm. The plates were stacked ensuring their separation using stainless steel spacers, and were preconditioned by immersion in water for 30 days at 20 °C.

After preconditioning, the plates were placed inside the FT chamber for 100, 200, and 300 cycles. The duration of each cycle (Figure 2) was 13 h and 25 min consisting of the thawing stage, with test material submerged in water for 8 h at 23 °C, and the freezing stage in a dry condition for 3 h at −20 °C, together with heating and cooling stages of 1.5 h to ensure isothermal conditions throughout the FT chamber, as well as inside the bulk of the composite plates. The water was drained from the FT chamber before the freezing stage and pumped back in at the end of the heating stage. The thermal response of the FT chamber and in the core of a dummy composite specimen was monitored with two thermocouples throughout a number of cycles; a full cycle is shown in Figure 2.

### 3.2. Experimental Characterisation

The properties of the composites were assessed before accelerated ageing, after the preconditioning, and after exposure to 100, 200, and 300 FT cycles. For each batch, the plates were cut in a computer numerical control (CNC) milling machine into individual test specimens of geometry defined according to relevant test standards for mechanical and thermomechanical tests, as shown in Figure 3, and then stored in water until the time of the experiment. During the mechanical tests, the strains developed in the materials were obtained with a video extensometer consisting of a *SONY FLIR BFS-U3-51S5M-C* camera, *Fujinon HF25SA-1* lens, and *MatchID* software (v2022.2) for data acquisition. For each series of mechanical tests, at least five specimens were tested, while for DMA two specimens were tested in each series. All the aged specimens were tested in wet conditions. A digital micrometre was used to measure the thickness and width of the specimens along their gauge length (a minimum of three measurements for each dimension were taken). In parallel to the FT-cycle exposure, the water absorption characteristics of the composites were also investigated in accelerated conditions, as detailed next.

The moisture absorption test was performed following a combination of recommendations found in three relevant standards (ISO 62, ASTM 570, and ASTM D5229). Square basalt–polymer composite specimens were used, and they were prepared by CNC milling with a 100 mm side and a nominal thickness of 4 mm. During specimen preparation, the surface and the edges were cleaned to remove any residue from CNC milling. Before immersion in water, the specimens were preconditioned by drying in a chamber at 35 °C and 30% RH for 24 h, and then kept in a dry chamber at 25 °C and 30% RH for 60 days until reaching a constant mass (mass variation below 0.1%). Subsequently, the specimens were immersed in water at 20 °C. The water uptake was measured throughout the specific periods of 1, 2, 4, 8, 14, 21, 30, and 60 days. For weighing, the specimens’ surface and edges were cleaned with lint-free cloths and the weight measurements were made immediately. The cleaning and measuring processes were completed within 1 min. For each thermal condition and material, three specimens were used.

The thermomechanical behaviour of the composite was assessed using a dynamic mechanical analyser from *TA instruments*, (Newcastle, DE, USA) model *Q800 TA*, following parts 1 and 5 in the ISO 6721 standard, using specimens with geometry of 60 mm by 10 mm. The amplitude of the strain and frequency were 15 µm and 1 Hz, respectively. The specimens were tested in the temperature range −30 to 150 °C, at a heating rate of 2 °C/min. T_g_ was measured from the onset of the storage modulus decay and from the peak of the tan(δ).

The tensile properties of the composites were determined following parts 1 and 4 in the ISO 527 standard. The geometry of the tensile specimens was 300 mm by 25 mm. A universal test machine (UTM) from Instron (Norwood, MA, USA), model 5982, with a load capacity of 100 kN, was used and tests were conducted at a cross-head speed of 2 mm/min.

The compressive strength of the composites was determined using a combined loading compression (CLC) test setup according to the ASTM 6641/D6641M standard [40]. The specimen geometry was 147 mm by 25 mm. The tests were conducted in the same UTM at a cross-head speed of 1.3 mm/min.

The in-plane shear (perpendicular to the main axis of the composite) properties of the composites were measured following standard ASTM D5379/D5379M, using specimens with dimensions 76 mm by 20 mm and comprising a V-notch at the central section. The tests were conducted in the same UTM at a cross-head speed of 2 mm/min.

The interlaminar shear strength of the composites was determined using the short beam method following the ISO 14130 standard [41]. The geometry of the specimens was 40 mm by 20 mm. The tests were conducted in the same UTM at a cross-head speed of 1 mm/min.

SEM observations were carried out on randomly selected samples extracted from the two composites. The samples were taken from failed tensile specimens, allowing direct access to strands of delaminated material without the need for additional cutting or processing (which could affect the samples and the resulting observations). SEM observations were made for (i) unaged specimens, (ii) specimens tested after preconditioning, and (iii) specimens tested after 300 FT cycles. A *Hitachi* (Tokyo, Japan) scanning electron microscope, model *S2400*, equipped with a SDD light elements energy-dispersive spectroscopy (EDS) Bruker detector (Bruker, Billerica, MA, USA) was used. The samples were placed on an aluminium stub using double-sided carbon tape and were prepared by sputter coating with a thin gold/palladium film in a *Quorum Technologies* coater (Sacramento, CA, USA), model *Q150T ES*.

## 4. Results and Discussion

### 4.1. Initial Characterisation

Figure 4 and Table 2 present the initial performance of unaged bio- and oil-based basalt fibre reinforced composites in terms of their mechanical properties in tension, compression, in-plane shear, interlaminar shear, and thermomechanical behaviour. The mechanical behaviour of both composites was very consistent and in line with the typical response from similar fibre–polymer composites. In most cases, the overall performance of the bio-composite was comparable to that of the oil-based counterpart. The average tensile strength, compressive strength, and apparent interlaminar shear strength of the bio-composite were around 10% lower than those of the oil-based one, while both presented similar tensile modulus (this property is fibre-dominated and the fibre architecture was identical). Slightly higher differences were observed for the in-plane shear modulus (also dependent on the matrix), which was 15% lower in the bio-composite compared to the conventional oil-composite. The in-plane shear strength of both composites was almost similar.

The storage modulus curves for both composites present a well-defined and typical development, with sharp drops at the glass transition region. Although the storage moduli of both composites are similar in the glassy state, in the rubbery plateau the oil-composite presented a lower value of storage modulus. T_g,onset_ (onset of decay of storage modulus) of both composites was almost similar. The tan(δ) curves for both types of composites also present a typical development, showing a well-defined peak at glass transition, without signs of secondary relaxations, indicating a high curing degree in both resins. A slightly higher T_g,tan(δ)_ was obtained for the bio-composite. Additionally, the lower loss modulus and tan(δ) curve peaks indicate that the bio-composite has lower capacity for energy dissipation, possibly resulting from higher residual fibre–matrix adhesion at temperatures above T_g_ compared to the oil-based composite.

### 4.2. Water Absorption

Figure 5 shows the water uptake in the bio- and oil-based composites at 20 °C (the preconditioning method prior to FT-cycle exposure) for up to 60 days. The water absorption is significantly higher in the bio-composite compared to the oil-based counterpart. Around 80% of the water uptake in the oil-composite occurred within 30 days of immersion, while in the bio-composite that figure was around 92% (both with respect to the 60 days of immersion), and moisture absorption after the preconditioning period was 0.46% and 1.73% in the oil-composite and the bio-composite, respectively.

The higher uptake of water in the bio-composite might be explained by the renewably sourced monomers, such as ISO and PDO, as well as the oil-based HEMA used in the bio-based UPR as a replacement for styrene [42]. These molecules have a hydroxyl group in their chemical structure that can form hydrogen bonds with water molecules. Unreacted volatiles (not measured), such as styrene, also might have contributed to the water absorption process.

### 4.3. Tensile Properties

As shown in Figure 6a and Table 3, the tensile strength of both bio-composite and oil-composite decreased moderately during preconditioning and 100 cycles of FT exposure, with reductions of around 4 to 16%. In the bio-composite, the reduction rate was higher compared to the oil-based counterpart. After 100 FT cycles, the tensile strength seems to have stabilised in the bio-composite, while in the oil-composite the decreasing trend continued until 300 cycles. The retention in both composites after 300 cycles was found to be 82% and 84% for the bio-composite and oil-composite, respectively. During the preconditioning period, the damage most likely occurred by moisture-induced phenomena, such as plasticisation, hydrolysis, and fibre–matrix debonding [43]; for this stage, the higher degradation levels experienced by the bio-composite are consistent with its higher water uptake compared to the oil-composite. Similar trends and mechanisms of tensile strength reduction have been reported in other studies [26,28,32].

Regarding tensile modulus (Figure 6b), the oil-composite was not affected by the immersion preconditioning, while the bio-composite exhibited a reduction of around 5%; however, this was later recovered after 200 FT cycles, and even for the oil-composite, the tensile modulus showed fluctuations of the same order of magnitude as the bio-composite throughout the FT assessment. After 300 FT cycles, the tensile modulus retention was around 95% in both composites. The small magnitude and lack of a clear trend in these variations can be explained by this property’s lower dependence on matrix properties, thus not clearly evidencing any changes that might occur in the polymer resins. These results agree well with those reported in other FT studies [26,28,32].

### 4.4. In-Plane Shear Properties

The in-plane shear strength, shown in Figure 6c and Table 3 for both composites, was found to be significantly affected during preconditioning, which caused reductions of around 40% and 20% in the bio-composite and oil-composite, respectively; once again, the worse performance of the bio-composite during this preconditioning stage is in line with its higher water uptake. However, property recovery was noticed in both materials during the FT cycles. After 300 FT cycles, the property retention in the bio-composite was 66%, while that in the conventional counterpart was 93%. Being a property where the resin matrix and fibre–matrix interaction play significant roles, the severe degradation in the in-plane shear strength of the bio-composite suggests that the physical and chemical damage mechanisms, such as plasticisation, hydrolysis, and fibre–matrix debonding, were more prevalent in the bio-based resin than in the conventional one.

The damage caused by the preconditioning period in the in-plane shear modulus (Figure 6d and Table 3) was also found to be severe, with reductions of 25% and 10% in the bio-composite and oil-composite, respectively. The reductions in this property, also very much dependent on the resin matrix (note that the composites used in the experiment only include reinforcement oriented at 0° and 90°), progressed for 100 FT cycles, with some recovery and stabilisation being observed for 200 FT cycles. After 300 FT cycles, the retention in the in-plane shear modulus was found to be around 82% and 94% in the bio-composite and oil-composite, respectively.

**Table 3 materials-16-05411-t003:** Average ($\bar{x}$ ), standard deviation (SD), coefficient of variation (CV), and retention (R) of mechanical properties and T_g_ after the preconditioning and the freeze–thaw-cycle exposure.

		Bio-Composite				Oil-Composite			
Property	Parameter	Preconditioning	100 Cycle	200 Cycle	300 Cycle	Preconditioning	100 Cycle	200 Cycle	300 Cycle
Tensile strength	$\bar{x}$ [MPa]	487.3	452.4	450.0	439.4	558.5	536.8	505.2	489.6
	SD	11.0	13.0	7.3	21.5	10.2	10.3	9.7	25.8
	CV [%]	2.3	2.9	1.6	4.9	1.8	1.9	1.9	5.3
	R [%]	91	84	84	82	96	92	87	84
Tensile modulus	$\bar{x}$ [GPa]	26.5	27.1	27.9	26.3	27.9	26.6	26.8	26.0
	SD	0.4	0.6	0.4	0.3	0.6	0.8	0.7	0.5
	CV [%]	1.8	2.4	1.3	1.0	2.2	2.9	2.4	2.0
	R [%]	95	97	100	95	100	96	96	94
In-plane shear strength	$\bar{x}$ [MPa]	26.6	26.2	27.5	29.0	33.5	38.9	36.5	39.9
	SD	0.4	1.0	0.8	1.4	1.6	1.6	1.0	0.5
	CV [%]	1.4	3.7	2.9	4.8	4.9	4.1	2.7	1.3
	R [%]	61	60	63	66	78	91	85	93
In-plane shear modulus	$\bar{x}$ [MPa]	2.0	1.9	2.3	2.2	2.8	2.5	2.6	2.9
	SD	0.2	0.2	0.2	0.1	0.1	0.3	0.2	0.2
	CV [%]	9.1	10.7	6.4	4.4	2.3	10.1	8.9	8.1
	R [%]	75	69	84	82	90	81	85	84
Compressive strength	$\bar{x}$ [MPa]	90.0	94.3	97.5	90.8	124.6	128.3	124.4	133.3
	SD	7.1	20.1	11.6	18.0	12.2	16.5	10.4	6.6
	CV [%]	7.9	21.3	11.9	19.9	9.8	12.8	8.4	5.0
	R [%]	63	66	68	63	79	82	79	85
Interlaminar shear strength	$\bar{x}$ [MPa]	9.1	11.3	11.9	11.3	>14.2	16.4	15.7	16.1
	SD	0.4	0.3	1.2	0.6	0.6	0.8	0.8	0.7
	CV [%]	4.1	3.0	10.3	5.1	4.0	4.7	4.8	4.3
	R [%]	58	72	76	70	81	94	90	92
T_g,onset_	[°C]	49.6	39.4	44.6	46.8	64.5	62.5	63.7	59.9
	R [%]	72	57	65	68	91	88	90	85
T_g,tan(δ)_	[°C]	73.5	76.8	76.6	77.0	87.6	89.0	89.0	85.7
	R [%]	75	78	78	78	93	95	95	91

**Figure 6 materials-16-05411-f006:**
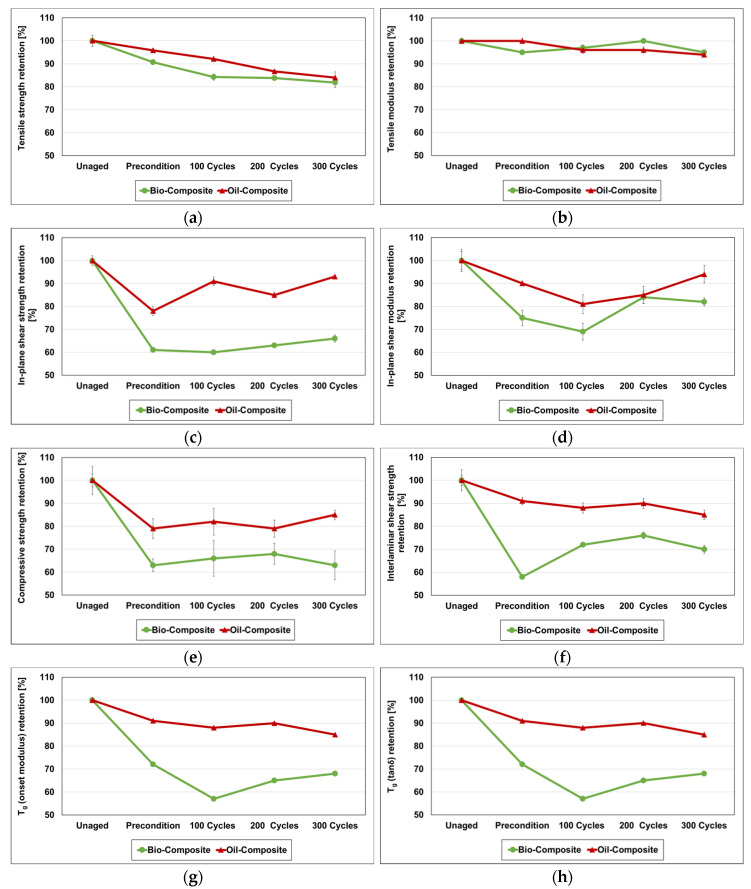
Property retention (in %, average values and error bars, where applicable, corresponding to one standard deviation): (**a**) tensile strength, (**b**) tensile modulus, (**c**) in-plane shear strength, (**d**) in-plane shear modulus, (**e**) compressive strength, (**f**) interlaminar shear strength, (**g**) T_g,onset_, and (**h**) T_g,tan(δ)_.

### 4.5. Compressive Strength

The degradation in compressive strength (Figure 6e and Table 3) followed a similar trend to that for other matrix-dominated properties, such as in-plane shear strength, presumably for the same reasons highlighted before. Severe degradation in compressive strength occurred in both types of composites due to preconditioning, with reductions of 37% and 21% in the bio-composite and oil-composite, respectively. However, during the FT cycles the compressive strength retention stabilised, showing only slight fluctuations. After 300 FT cycles, the property retention reached around 63% and 85% in the bio-composite and oil-composite, respectively. Most of the degradation in compressive strength occurred during the preconditioning period, while FT cycles caused insignificant damage. It is also worth mentioning that comparatively high variability has been noticed in the compressive strength properties of both composites, possibly stemming from this property’s higher susceptibility to local defects and variations in fibre alignment [44].

### 4.6. Interlaminar Shear Strength (ILSS)

The ILSS provides information regarding the adhesion between laminas in the composite material, as well as the fibre–matrix adhesion. The ILSS of the bio-composite decreased significantly during the water immersion preconditioning, showing a reduction of 32% compared to the unaged material (Figure 6f and Table 3). In the conventional composite, the effect of preconditioning was much lower, with a 9% reduction. The ILSS reduction during the preconditioning of the composite can be attributed to the water-associated damage mechanisms discussed earlier, with the more severe damage in the bio-composite being consistent with the higher hydrophilicity of the bio-based resin used in the bio-composite. However, in the bio-composite, the damage from the preconditioning recovered significantly after 100 FT cycles (variation of +14%), followed by a property stabilisation up to 300 FT cycles. To some extent, the drying phase in the FT cycles might have contributed to the recovery of ILSS of the bio-composite. Additionally, post-curing in the freezing condition contributed to the property enhancement in the bio-composite. In the oil-based composite, permanent damage was observed, with the ILSS not recovering during the FT cycles, but rather remaining approximately constant. After 300 FT cycles, the retention of ILSS was 70% and 80% in bio- and oil-composites, respectively, which is in line with other studies [28].

### 4.7. Dynamic Mechanical Analysis (DMA)

The DMA allows assessing the glass transition of the composite, a matrix-dominated process resulting from changes in its thermomechanical response with increasing temperature, from a glassy, brittle state to a rubbery amorphous state; while it occurs gradually over a given temperature range, this process is typically characterised by a specific temperature, denoted as glass transition temperature (T_g_), which can be defined from different DMA parameters. In this study, as mentioned, the T_g_ was investigated from the onset of the decay of the storage modulus curve (T_g,onset_), as well as from the peak of the tan(δ) (T_g,tan(δ)_). The storage modulus and tan(δ) curves resulting from DMA of the unaged and aged samples are shown in Figure 7.

Similar to other matrix-dominated properties, DMA results indicate the occurrence of severe degradation during the preconditioning stage, caused by moisture-induced damage mechanisms, specifically in the bio-composite, as shown in Figure 7a and Figure 6g, and Table 3. This 30 day period of water immersion caused a very significant T_g,onset_ reduction in the bio-composite (28%), much higher than that experienced by the oil-composite (9%). The trend of reduction in the bio-composite continued, reaching a 57% retention of T_g,onset_ after 100 FT cycles, which then started to be recovered to some extent (in line with most matrix-dominated mechanical properties). In contrast, the T_g,onset_ of the oil-composite seemed negligibly affected during the FT cycles, resulting in a stabilisation of that property after the preconditioning stage. Water immersion and absorption resulted in resin plasticisation, which in turn reduced the T_g,onset_ of the materials. This reversible mechanism may have been more prominent in the aged bio-based composite, as suggested by the ‘shoulder’ of the damping curves in Figure 7a for temperatures around 100 °C, which indicates the potential evaporation of water from the DMA sample. Furthermore, hydrolysis reactions affecting the long polymer chains as well as the side chains are likely to have caused the leaching of small molecular weight segments, thus weakening the polymer matrix. During the FT cycles, the expansion of absorbed water molecules at the freezing stage and the contraction at the thawing stage may also have resulted in some micro-cracking in the polymer matrix of the composite, also contributing to the degradation of T_g,onset_. However, such degradation during FT cycles was offset to some extent by the embrittlement of the polymer matrices as well as post-curing during FT, as suggested by the relative reduction in the magnitude of the tan(δ) peaks, particularly for the bio-based composite. These phenomena might justify some of the property recovery during FT cycles. The retention of T_g,onset_ was found to be 68% and 85% after 300 FT cycles for the bio-composite and oil-composite, respectively, mostly resulting from a significant level of permanent damage occurring during preconditioning, particularly in the bio-composite. Other studies [31,45] have also found insignificant T_g_ degradation (3 to 5%) in GFRP composites due to FT exposure.

Regarding T_g,tan(δ)_, both composites showed (Figure 6h and Figure 7b and Table 3) less degradation comparing to the degradation in T_g,onset_. In the bio-composite, after preconditioning, a degradation of 25% was observed and the property stabilised with 78% retention throughout the 300 FT cycles. In the oil-composite, the preconditioning period caused a reduction of 7%, followed by stabilisation with property retention of 95% for up to 200 FT cycles, with a slight reduction of 4% after 300 FT cycles. Similar moisture-related degradation mechanisms justify the initial degradation during preconditioning, as discussed earlier, albeit with a lower magnitude compared to the bio-composite. However, the FT condition had only minor effects on T_g,tan(δ)_.

### 4.8. Scanning Electron Microscopy (SEM)

Figure 8 shows the SEM micrographs of the bio- and oil-composites in the unaged, preconditioned, and after 300 FT-cycle conditions.

The SEM observations were made for multiple locations on the fracture surface of the samples extracted from failed tensile specimens. From these images, it was possible to observe both cohesive failure in the polymer resin as well as failure in the fibre–matrix interface for both materials. For the unaged condition, the SEM observations did not suggest any significant differences between the two types of composites (lower magnification images also did not show relevant differences). Comparing the unaged and preconditioned specimens, it was possible to observe a reduction in the volume and integrity of the resin fragments adherent to the basalt fibres, particularly for the bio-composite, suggesting that both the fibre–matrix adhesion and the resin strength were affected by water immersion. Between the preconditioned and 300 FT-cycle samples, no significant differences were noticeable, suggesting that most of the degradation occurred from the moisture ingress and its associated chemical and physical degradation mechanisms during the preconditioning phase. These observations, although obtained in very specific locations, are in good agreement with the experimental results previously presented and discussed.

## 5. Conclusions

This paper presented a study of the wet FT durability of a bio-composite produced by vacuum infusion with bio-based unsaturated polyester resin reinforced with basalt fibres, comparing it to an equivalent composite produced using a conventional oil-based resin. Both composites were preconditioned in water immersion for 30 days and subsequently exposed to up to 300 FT cycles.

The initial (unaged) properties of both composites were found to be overall comparable. However, the moisture uptake by the bio-composite in water immersion at 20 °C after 60 days was three times higher than in the oil-composite, highlighting the significantly higher hydrophilicity of the bio-based UP resin; this was attributed to the hydroxyl groups in the chemical structures of some of the monomers used, such as PDO, isosorbide, and HEMA. This had direct impacts on the generally higher degradation in mechanical and thermomechanical properties found for the bio-composite after preconditioning in water for 30 days. A notable exception to this was found in the tensile modulus of both composites, which showed very little degradation after preconditioning of FT cycling, reflecting the fibre-dominated character of this property.

The gap in property degradation between the bio-composite and the oil-composite was typically wider for matrix-dominated properties. Severe degradation was observed in the ILSS of the bio-composite, with around 42% reduction during the preconditioning period followed by a slight property recovery (14%) and stabilisation during FT exposure; in the oil-based counterpart, ILLS was found to be insignificantly affected by the FT conditioning, with most of the degradation (10%) occurring during the preconditioning period, followed by property stabilisation with slight fluctuations. This finding was also supported by SEM observations of changes in the patterns of resin fragments adherent to the basalt fibres occurring most visibly between unaged and preconditioned specimens.

During the FT cycles, property recovery was generally noticed in both composites, with the bio-composite generally showing higher recovery, possibly indicating the occurrence of post-curing and embrittlement of the bio-based resin, both of which are supported by the lowering of the magnitude of the damping curve peaks in DMA of aged samples.

Overall, this study highlighted the inferior durability performance of the bio-based basalt fibre reinforced composite under FT exposure, and this was mostly attributed to the worse ability of the bio-based resin in resisting moisture-induced degradation compared to the conventional oil-based counterpart. Conversely, both composites did not show significant additional degradation stemming from FT cycling. Despite this worse performance from the bio-based resin, these results suggest that its use with basalt fibre reinforcement is still viable and that to improve the hydrolytic resistance of the bio-based resin it is essential to fine-tune its composition in order to limit its hydrophilicity.

## Figures and Tables

**Figure 1 materials-16-05411-f001:**
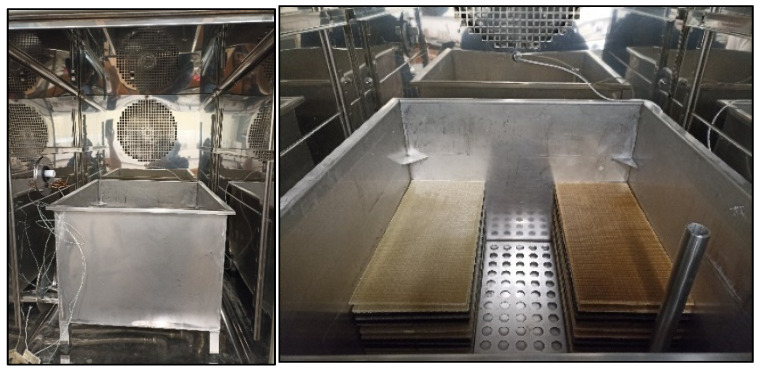
Freeze–thaw chamber and composite stacking.

**Figure 2 materials-16-05411-f002:**
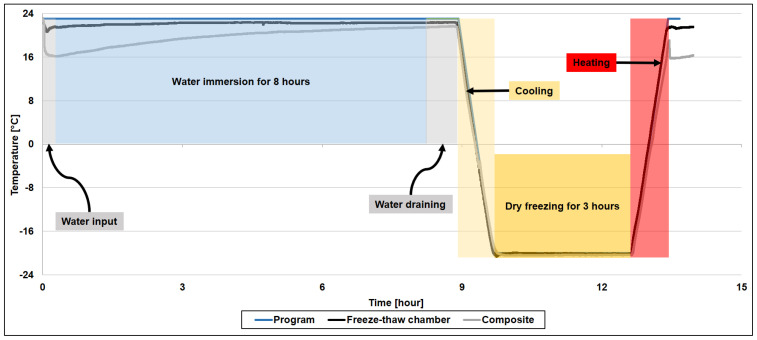
Thermal response of the freeze–thaw chamber and of a dummy specimen during one complete cycle.

**Figure 3 materials-16-05411-f003:**
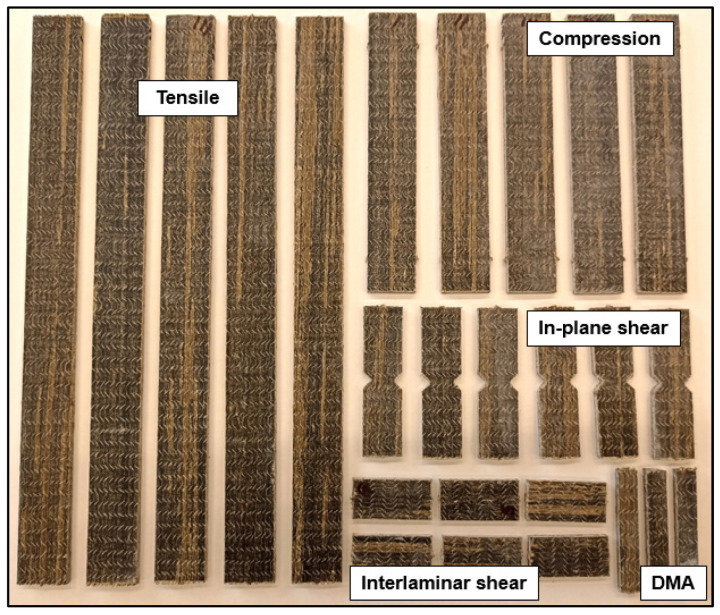
Composite specimens for mechanical and thermomechanical tests.

**Figure 4 materials-16-05411-f004:**
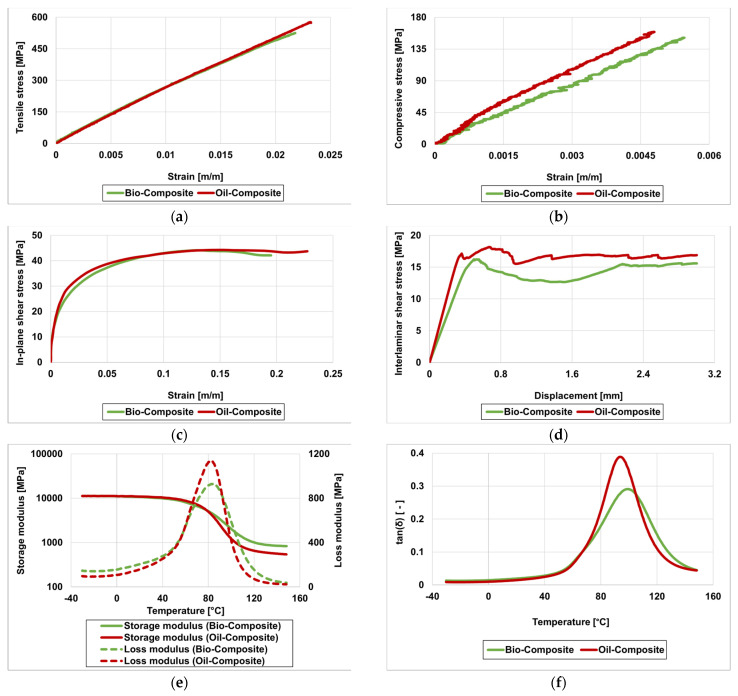
Initial characterisation tests of bio-composite and oil-composite (one representative curve for each material): (**a**) tensile, (**b**) compressive, (**c**) in-plane shear, (**d**) interlaminar shear, (**e**) DMA—storage and loss modulus, and (**f**) DMA—tan(δ).

**Figure 5 materials-16-05411-f005:**
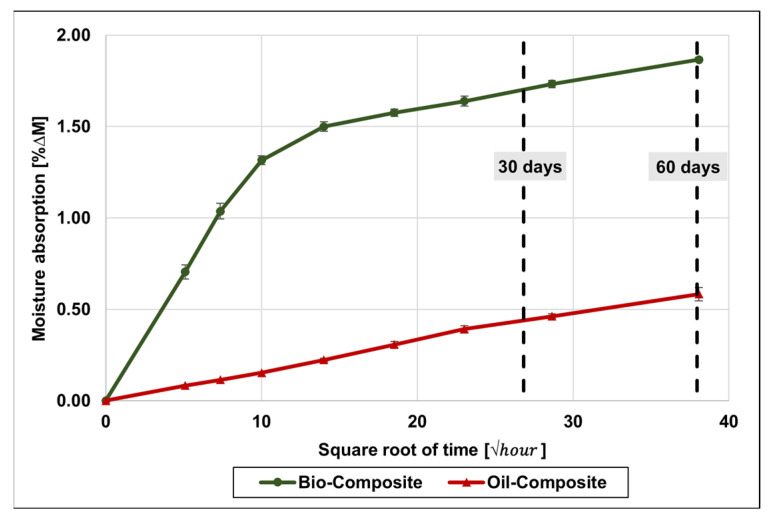
Moisture absorption profile for bio-composite and oil-composite at 20 °C, including standard deviation.

**Figure 7 materials-16-05411-f007:**
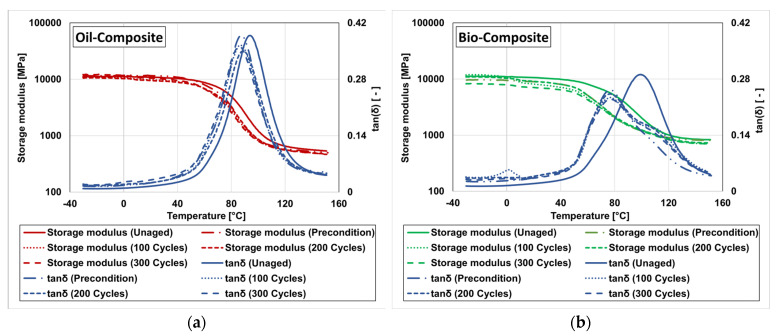
Storage modulus and tan(δ) of (**a**) bio-composite and (**b**) oil-composite before ageing, and after preconditioning and exposure to different FT cycles.

**Figure 8 materials-16-05411-f008:**
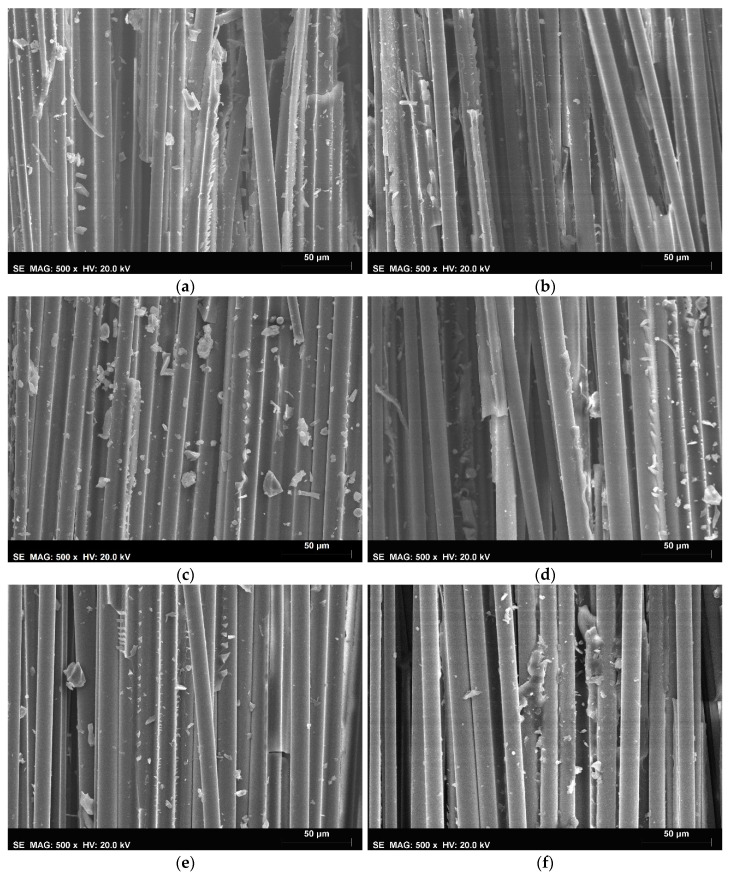
SEM micrographs (500× magnification) of the basalt fibre composites: (**a**) unaged bio-composite, (**b**) unaged oil-composite, (**c**) bio-composite after preconditioning, (**d**) oil-composite after preconditioning, (**e**) bio-composite after 300 FT cycles, and (**f**) oil-composite after 300 FT cycles.

**Table 1 materials-16-05411-t001:** Mechanical and thermomechanical properties of bio-based and oil-based [21] UPRs.

Properties		Bio-UPR	Oil-UPR	Standard (No. of Specimens)
T_g_	Onset E’ modulus [°C]	70.6	75.3	ISO 6721-11 [37] (2)
	tan(δ) [°C]	103.6	102.4	
Tension	σ_t_ [MPa]	43.5 ± 3.4 (7.8%)	45.3 ± 3.3 (7.3%)	ISO 527-4 [38] (10)
	E_t_ [GPa]	3.3 ± 0.03 (0.9%)	3.3 ± 0.1 (3.4%)	
	ε_t_ (m/m) [%]	1.6 ± 0.1 (8.8%)	1.7 ± 0.2 (11.7%)	
In-plane shear	σ_12_ [MPa]	49.0 ± 3.4 (6.9%)	54.1 ± 2.5 (4.7%)	ASTM D5379/D5379M [39] (8)
	G_12_ [GPa]	1.6 ± 0.1 (8.4%)	1.3 ± 0.1 (10.8%)	

**Table 2 materials-16-05411-t002:** Initial properties of bio-based composite and oil-based composite (average ± standard deviation values when applicable, coefficient of variation %).

Properties	Parameter	Bio-Composite	Oil-Composite
T_g_	Onset E’ [°C]	69.2	70.9
	tan(δ) [°C]	98.2	93.8
Tension	σ_t_ [MPa]	537.1 ± 23.6 (4.4%)	583.0 ± 10.7 (1.8%)
	E_t_ [GPa]	27.8 ± 0.3 (1.1%)	27.8 ± 0.6 (2.2%)
Compression	σ_c_ [MPa]	143.3 ± 7.2 (5.0%)	157.1 ± 17.2 (10.9%)
In-plane shear	τ_12_ [MPa]	43.8 ± 0.7 (1.6%)	42.8 ± 1.8 (4.2%)
	G_12_ [GPa]	2.7 ± 0.2 (7.9%)	3.1 ± 0.3 (9.2%)
Interlaminar shear	σ_sbs_ [MPa]	15.7 ± 1.2 (7.6%)	17.5 ± 1.6 (9.2%)

## Data Availability

Not applicable.

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
