# Peer review of "Freeze–Thaw Durability of Basalt Fibre Reinforced Bio-Based Unsaturated Polyester Composite"

_materials, 2023, doi:10.3390/ma16155411_

Round 1
Reviewer 1 Report
The authors report a study on the Freeze-thaw durability of basalt fibre reinforced bio-based unsaturated polyester composite. In particular they compare the mechanical behavior of innovative bio-based unsaturated polyester with a conventional oil-based resin. The manuscript is interesting, however I that should be improved to be accepted in Materials.
- Since bio-based is also used in the title, please clarify if he “new polyester” is fully or partially biobased? Please define the of unsaturated and they amount in the final polymer.
- The mechanical parameters reported in Table 1 are those determined for the Bio-UPR of ref 21. Correct? The polyester prepared in this work has different HEMA/styrene ratio and different properties.
- The authors should also study the properties of the two polymers without fillers.
- The authors report that “Unreacted volatiles, such as styrene, also might have contributed to the water absorption” I think that unreacted monomer are not desirable. Which is the amount of unreacted monomers?
- It is very hard to see differences in the SEM images reported in the manuscript. I only see fiber details at high magnification.
- Please add more details, if possible, on the composite preparation and filler loading.
Author Response
Reviewer #1
General remarks: The authors report a study on the Freeze-thaw durability of basalt fibre reinforced bio-based unsaturated polyester composite. In particular they compare the mechanical behavior of innovative bio-based unsaturated polyester with a conventional oil-based resin. The manuscript is interesting, however I that should be improved to be accepted in Materials.
Reply: The authors would like to thank the reviewer for assessing our paper and the comments provided, which contributed to improve the manuscript. Below, please find our responses to the specific comments provided and the actions prompted by those remarks.
Remark #1: Since bio-based is also used in the title, please clarify if he “new polyester” is fully or partially biobased? Please define the of unsaturated and they amount in the final polymer.
Reply: As referred in line 166 of the (original) manuscript, the unsaturated polyester resin is “partially bio-based”. As detailed in [21], except for phthalic anhydride, all other monomers used in the synthesis of the polyester prepolymer were obtained from renewable resources, resulting in bio-based carbon and mass contents of 78.6% and 84.1%, respectively (determined as per ISO 16620).
We now indicated the bio-based content in the revised version.
Action 1: In the revised version, in section 2, paragraph 1, we changed “in the orthopthalic polyester pre-polymer chain” to “in the orthopthalic polyester pre-polymer chain (bio-based mass content of 84.1%, as per ISO 16620)”.
Action 2: In the revised version, in section 2, paragraph 1, we changed “followed by mixing with reactive diluents (RDs) (50 parts of partially bio-based unsaturated polyester pre-polymer to 50 parts of RDs).” to “followed by mixing with reactive diluents (RDs) (50 parts of partially bio-based unsaturated polyester pre-polymer to 50 parts of RDs), which resulted in an overall bio-based mass content of 42%”.
Remark #2: The mechanical parameters reported in Table 1 are those determined for the Bio-UPR of ref 21. Correct? The polyester prepared in this work has different HEMA/styrene ratio and different properties.
Reply: No, the mechanical properties listed in Table 1 are those determined for the polyester prepared in this work (which, as noted by the reviewer, has a different HEMA:styrene ratio (40:60 vs. 50:50 in the previous work). The reference to the previous work [21] was misleading as it only applies to the oil-based resin. We now clarified this point.
Action: The caption of Table 1 was revised and it now reads as follows: “Mechanical and thermomechanical properties of bio-based and oil-based [21] UPRs.”.
Remark #3: The authors should also study the properties of the two polymers without fillers.
Reply: None of the polymers (bio-based or oil-based) contain any fillers. The paper made no mention to fillers. In the revised version, we now clarified that no fillers were used in the polymers.
Action: In section 2, at the end of paragraphs 1 and 2, we added, respectively “No additives were used.” and “(…) and it also contained no additives”.
Remark #4: The authors report that “Unreacted volatiles, such as styrene, also might have contributed to the water absorption” I think that unreacted monomer are not desirable. Which is the amount of unreacted monomers?
Reply: We did not measure the amount of unreacted volatiles. We only mentioned this (potential) effect as one of the possible (not confirmed) causes for the waster absorption process. In the revised version we clarified this point.
Action: We changed “Unreacted volatiles” to “Unreacted volatiles (not measured)”.
Remark #5: It is very hard to see differences in the SEM images reported in the manuscript. I only see fiber details at high magnification.
Reply: Yes, SEM images did not show significant differences between both types of BFRP composites (this was mentioned in the original version of the manuscript). Lower magnifications also did not show significant differences. We clarified this point the revised version.
Action: The sentence “(…) the SEM observations did not suggest any significant differences between the two types of composites.” was changed to “(…) the SEM observations did not suggest any significant differences between the two types of composites (lower magnifications also did not show relevant differences)”.
Remark #6: Please add more details, if possible, on the composite preparation and filler loading.
Reply: As explained in the reply to remark #3, the polymeric matrix did not contain any filler. We now provided a few more details about the composite preparation.
Action: The following new paragraph was added at the end of section 2, it includes some of the information previously included in “Basalt fibre mats” and additional information about the composite preparation.
“Production of composite plates: Both bio-based and oil-based UPR were used to produce basalt-polymer composite plates using the vacuum infusion technique, which comprises the following three main stages: (i) preparation - stacking of the different mat layers over a metallic moulding plate (size of 1.25 m x 1.25 m), together with infusion meshes, peel ply and breather, and application of a plastic film (as well as other infusion accessories) and vacuum (pressure of about 40 mbar) for approximately 24 hours; (ii) resin infusion, with visual observation of resin percolation and temperature increase (the vacuum system was connected for a period of 24 hours); and (iii) demoulding. The composite laminates, with 64 cm x 75 cm, were cured at room temperature for 48 hours and then post-cured at 100 °C for 4 hours. The nominal thickness of the plates was 4 mm, and the fibre mass content of the laminates was around 65%.”

Reviewer 2 Report
The article presents interesting solutions, especially in the environmental aspect. However, I have a few remarks:
- Fig. 3 shows the defects to which the Authors did not respond.
- no definition of test equipment, i.e. type, manufacturer
- why is the compression speed 1.3 mm/min?
- was there any rule of sputtering alternatively gold/palladium for samples for microscopic examination?
In general, I believe that the research methodology, discussion of results, and conclusions can be extended and improved.
Author Response
Reviewer #2
General Remarks: The article presents interesting solutions, especially in the environmental aspect. However, I have a few remarks:
Reply: The authors would like to thank the reviewer for assessing our paper and the comments provided, which contributed to improve the manuscript. Below, please find our responses to the specific comments provided and the actions prompted by those remarks.
Remark #1: Fig. 3 shows the defects to which the Authors did not respond.
Reply: Figure 3 illustrates the geometry of the composite specimens used in the different mechanical and thermo-mechanical tests. It does not illustrate any defects; in fact, visual observations and geometric measurements did not show any relevant defects in test specimens.
Action: None.
Remark #2: no definition of test equipment, i.e. type, manufacturer
Reply: We now included additional information about the main test equipment used in the experimental tests.
Action 1: About DMA, “The thermomechanical behaviour of the composite was assessed” was changed to “The thermomechanical behaviour of the composite was assessed using a dynamic mechanical analyzer from ‘TA instruments’, model ‘Q800 TA’.”
Action 2: About Tensile tests, “A universal test machine (UTM) (…)” was changed to “A universal test machine (UTM) from ‘Instron’, model ‘5982’, (…).”
Action 3: About Compression tests, “The test was conducted in a UTM (…)” was changed to “The test was conducted in the same UTM (…)”.
Action 4: About In-plane shear tests, “(…) comprising a V-notch at the central section, tested at a cross-head speed of 2 mm/min.” was changed to “comprising a V-notch at the central section. Tests were conducted in the same UTM at a cross-head speed of 2 mm/min”.
Action 5: About Interlaminar shear, “(…) 40 mm by 20 mm and the tests were conducted at a rate of 1 mm/min.” was changed to “(…) 40 mm by 20 mm. Tests were conducted in the same UTM at a rate of 1 mm/min”.
Remark #3: why is the compression speed 1.3 mm/min?
Reply: This test speed is the one recommended in the test standard used for these specific tests (ASTM D6641/D6641 M)
Action: None.
Remark #4: was there any rule of sputtering alternatively gold/palladium for samples for microscopic examination?
Reply: Gold and palladium were not used alternatively; the samples were prepared with a gold/palladium alloy, which is an industry standard coating for SEM.
Action: None.
Remark #5: In general, I believe that the research methodology, discussion of results, and conclusions can be extended and improved.
Reply/Action: We made an effort to improve the description of the methods (see also reply/action prompted by Remark 2) and also the discussion of results and conclusions. The reviewer did not refer specific aspects that should be improved, but we did an effort to extend the discussion. All changes are made with track changes.

Round 2
Reviewer 1 Report
The authors provided an improved version of the manuscript. The revised versione can now be accepted to be published on Materials.
Author Response
Reviewer #1
Remark #1: The authors provided an improved version of the manuscript. The revised versione can now be accepted to be published on Materials
Reply: The authors thank the reviewer for his assessment and recommendation.
Reviewer 2 Report
The article lacks an assessment of the impact of manufacturing technology on the sample's quality. Fig. 3 shows samples with obvious defects. In my opinion, defects were not taken into account in the analysis of the results.
Author Response
Reviewer #2
Remark #1: The article lacks an assessment of the impact of manufacturing technology on the sample's quality. Fig. 3 shows samples with obvious defects. In my opinion, defects were not taken into account in the analysis of the results.
Reply: As mentioned in the previous response, the specimens presented no defects stemming from their production method. The differences in colour that can be observed on the surface of the specimens originate from the reflection of light from the different strands of unidirectional basalt fabric, for which even slight differences in the angle of incident light produced a non-uniform appearance even in the dry mats. The figures below illustrate this effect. These fibres were supplied by Basaltex and used as received.
Action: None.
